# Chimeric Vesicular Stomatitis Virus Bearing Western Equine Encephalitis Virus Envelope Proteins E2-E1 Is a Suitable Surrogate for Western Equine Encephalitis Virus in a Plaque Reduction Neutralization Test

**DOI:** 10.3390/v17081067

**Published:** 2025-07-31

**Authors:** Kerri L. Miazgowicz, Bailey E. Maloney, Melinda A. Brindley, Mattie Cassaday, Raegan J. Petch, Paul Bates, Aaron C. Brault, Amanda E. Calvert

**Affiliations:** 1Arboviral Diseases Branch, Division of Vector-Borne Diseases, U.S. Centers for Disease Control and Prevention, Fort Collins, CO 80521, USAzlu5@cdc.gov (A.C.B.); 2Department of Infectious Diseases, Department of Population Health, College of Veterinary Medicine, University of Georgia, Athens, GA 30602, USA; mbrindle@uga.edu; 3Department of Microbiology, Perelman School of Medicine, University of Pennsylvania, Philadelphia, PA 19104, USA; rpetch@upenn.edu (R.J.P.); pbates@pennmedicine.upenn.edu (P.B.)

**Keywords:** plaque reduction neutralization test, serology, diagnostics, western equine encephalitis, vesicular stomatitis virus chimera, alphavirus, mosquito-borne viruses

## Abstract

In December 2023, infections of western equine encephalitis virus (WEEV) within Argentina were reported to the World Health Organization (WHO). By April 2024, more than 250 human infections, 12 of which were fatal, and 2500 equine infections were identified in South America. Laboratory diagnosis and surveillance in affected countries were hindered by a lack of facilities equipped with BSL-3 laboratories, as confirmatory serodiagnosis for WEEV requires live virus in the plaque reduction neutralization test (PRNT). To expand serodiagnosis for WEEV in the Americas, we developed a virus chimera composed of vesicular stomatitis virus (VSV) engineered to display the E2-E1 glycoproteins of WEEV (VSV/WEEV) in place of the VSV glycoprotein (G). PRNT_90_ and IC_90_ values of parental WEEV and VSV/WEEV were analogous using sera collected from mice, horses, and chickens. VSV/WEEV rapidly formed plaques with clear borders and reduced the assay readout time by approximately 8 h compared to the parental virus. Overall, we demonstrate that chimeric VSV/WEEV is a suitable surrogate for WEEV in a diagnostic PRNT. Use of chimeric VSV/WEEV in place of authentic WEEV will dramatically expand testing capacity by enabling PRNTs to be performed at BSL-2 containment, while simultaneously decreasing the health risk to testing personnel.

## 1. Introduction

From November 2023 to April 2024, a reemergence of Western equine encephalitis (WEE) in South America (Argentina, Brazil, and Uruguay) caused more than 250 recognized human infections, of which 12 were fatal, concurrent with equine infections exceeding 2500 [1,2,3,4]. In humans, WEE is often asymptomatic [5]. Symptomatic cases typically manifest as mild febrile illness; however, severe cases include neuroinvasive disease with case fatality rates ranging between 3 and 15% [6,7,8,9,10,11]. In horses, WEE has an elevated case fatality rate up to 40% with sequelae of weight loss, lethargy, and indicators of central nervous system involvement such as weakness, tremors, convulsions, and paralysis [12]. While both eastern equine encephalitis (EEE) and Venezuelan equine encephalitis (VEE) cause similar disease, the causative agents of each disease have distinct geographic distributions [13,14].

Western equine encephalitis virus (WEEV), the etiological agent of WEE, is an arbovirus in the genus *Alphavirus* (family *Togaviridae*) containing a single-stranded positive-sense RNA genome [15]. The WEEV genome consists of two open reading frames, which encode for four non-structural proteins (nsP1–4) and five structural proteins (C, E3, E2, 6K/TF, and E1) [16]. The structural proteins are assembled in a virion with T = 4 icosahedral symmetry, presenting E2-E1 heterodimers arranged as 80 trimeric spikes embedded in the lipid envelope [17].

WEEV transmission occurs through mosquito vectors (predominantly *Culex tarsalis*) in both enzootic cycles involving passerine birds and mammals, and epizootics involving larger mammals such as equines and humans who serve as dead-end hosts due to low-level transient viremia [18,19]. Epizootics of WEEV have been detected in the Americas sporadically since its isolation in 1930 [20]. However, a decline in the circulation and virulence of WEEV after the 1940s has been described [21,22,23], with no human cases of WEE reported in the US since 1999 [24]. Several veterinary vaccines for horses, such as Prestige 5 (Merck) and Core EQ Innovator (Zoetis), are available, with annual vaccination recommended in North America by the American Association of Equine Practitioners (AAEP). In January of 2024, Argentina reinstated a mandate requiring equine EEEV and WEEV vaccinations following an 8-year gap from a previous mandate effective from 2005 to 2016 (SENASA Resolution No. 521/16). Human vaccines for WEEV are under development [25,26,27,28], but are not currently available. Thus, mitigation strategies for human infection rely on mosquito vector population control and personal protection to prevent exposure to mosquito bites.

Diagnosis of WEE through direct detection of viral nucleic acid amplification tests (NAATs) from patient specimens is challenging due to the transitory and low-level viremia characteristic of many arboviruses [29]. Indirect immunoassays include an IgM antibody capture enzyme-linked immunosorbent assay (MAC-ELISA) for the detection of anti-WEEV IgM in diagnostic specimens [30]. Non-human primates exposed to WEEV developed detectable anti-WEEV IgM responses starting at day 4 and peaking at day 12 [29]; however, human anti-WEEV IgM could be detected as early as 1 day post symptom onset and peaked within the first two weeks for the examined cohort [31,32]. Furthermore, anti-WEEV IgM may persist for extended durations (months), underpinning the importance of testing for a 4-fold increase between acute and convalescent paired samples whenever possible [31]. Due to potential cross-reactivity associated with testing methods such as the MAC-ELISA, which does not require the use of live virus, confirmatory testing is performed with the highly specific plaque reduction neutralization test (PRNT) for detection of anti-WEEV neutralizing antibodies (NAbs). In a PRNT, an infectious virus is mixed with serum prior to inoculating a cellular monolayer. Presence of anti-WEEV NAb prevents virus-induced cell death, which is observed as plaque formation. WEEV PRNT requires infectious virus and biosafety level 3 (BSL-3) facilities [33].

The Pan American Health Organization (PAHO) published a public health risk assessment of WEEV in the Americas, emphasizing a high likelihood of underreporting during the 2023–2024 outbreak due to limited systematic surveillance and diagnostic capacity in affected countries [4]. These factors contributed to a ‘Moderate’ designation for risk of insufficient prevention and control capacity in the region [4]. Chimeric arboviruses have supported diagnostic and surveillance efforts by lowering the biosafety requirements and improving the safety profile for laboratory personnel conducting PRNTs [34,35]. Chimeric viruses are typically designed to contain antigens that confer equivalent recognition to the authentic virus in immunoassays but lack the pathogenicity associated with wildtype (WT) virus. For example, ChimeriVax chimeras developed for Dengue (DENV, serotypes 1–4) and Japanese encephalitis virus (JEV) were found to be suitable diagnostic alternatives in PRNT [34]. Similarly, chimeras developed for New World alphaviruses, Eastern and Venezuelan equine encephalitis virus (EEEV and VEEV, respectively), using Sindbis virus (SINV) as the viral core, were also found to be safe surrogates for the WT virus in PRNT [35,36].

To expand serodiagnosis for WEE, we developed replication-competent recombinant vesicular stomatitis viruses (rVSV) displaying the E2 and E1 envelope proteins of WEEV (Imperial Valley strain ‘Imp-181’ or McMillan) in place of its native glycoprotein (VSV G). VSV is a single-stranded negative-sense RNA virus of the family *Rhabodviridae* (genus *Vesiculovirus*). VSV chimeras are widely investigated as a vaccine platform for viral pathogens, including highly pathogenic agents such as Ebola virus (EBOV) (ERVEBO [37]), Lassa virus (LASV) [38], Nipah virus (NiV) [39], and Marburg virus (MARV) [40,41]. Furthermore, VSV-based vaccine vectors for both New World encephalitic [42] and Old World arthritogenic [43,44] alphaviruses have demonstrated protective efficacy in mouse models. VSV-based chimeras are attractive for use in serodiagnosis as VSV virions tolerate the incorporation of heterologous proteins and rapidly replicate to high titers. We found chimeric VSV/WEEV generated for both Group B3 strain Imp-181 and Group A strain McMillan performed equivalently to parental WEEV in PRNT. Use of chimeric VSV/WEEV as a surrogate for WT virus in PRNT lowers biosafety requirements for confirmatory diagnostics to BSL-2 containment and dramatically enhances arbovirus diagnostic testing capacity in public health laboratories.

## 2. Materials and Methods

### 2.1. Cells, Wildtype Viruses, and Immune Sera

Baby hamster kidney (BHK) cells stably expressing T7 polymerase (BSR T7/5 [45]) were a kind gift from Dr. Buchholz, with Vero cells and HEK293T cells sourced from the CDC Diagnostic Reagent Laboratory (DRT). Both BSR T7/5 and Vero cells were maintained in Dulbecco’s Modified Eagle Medium (DMEM) supplemented with 5% (*v*/*v*) fetal bovine serum (FBS), whereas HEK293T cells were cultured in DMEM supplemented with 10–15% (*v*/*v*) FBS. All cells were cultured at 37 °C with 5% CO_2_.

Parental WEEV strain Imp-181 was recovered from an infectious WEEV clone ‘MA2’ as previously described [46]. Briefly, the genome of WEEV (strain Imp-181, isolated from a *Cx. tarsalis* mosquito in 2005) was reconstructed from viral cDNAs and cloned into a pM1 vector using unique restriction sites. Virus recovery was performed by electroporation of in vitro transcribed RNA from the infectious WEEV clone into BHK-21 cells. Wildtype WEEV strain McMillan (human isolate collected in 1941) and all serum specimens evaluated (Appendix B Table A1) were acquired from the Center for Disease Control and Prevention Arbovirus Reference Collection (Fort Collins, CO, USA) except the positive anti-WEEV chicken specimen, which was supplied by the California Department of Public Health.

### 2.2. Generation and Recovery of VSV Chimeras

A replication-competent recombinant VSV particle (rVSV) bearing the E2 and E1 WEEV glycoproteins instead of its native glycoprotein (referred to within this study as ‘VSV/WEEV’) was generated by HiFi assembly (New England Biolabs, Ipswich, MA, USA) of the full-length VSV genome (serogroup Indiana, composed from strains Mudd-Summers and San Juan) from synthetic DNA fragments (Twist Bioscience, South San Francisco, CA, USA) designed with a gene placeholder (‘ATG’-‘TAG’) in place of the VSV G gene. The full-length clone design also included an extra VSV coding unit inserted at position 5, incorporating a gene placeholder (‘ATG-‘TAG’) to facilitate the easy insertion of reporter genes. The WEEV (strain Imp-181) envelope polyprotein (E3-E2-6K/TF-E1) was PCR amplified from the ‘MA2’ infectious clone plasmid and cloned into the glycoprotein coding unit placeholder with HiFi assembly. Likewise, the envelope polyprotein from WEEV (McMillan) was synthesized and PCR amplified to contain overlaps to the full-length clone prior to HiFi assembly. As a control, chimeric VSV/VSV was generated using the same approach by cloning synthesized VSV-G (Genbank ID:M35214.1) into the placeholder at position 4. The extremely bright green fluorescent protein ZsGreen (ZsG) was amplified from synthetic DNA and cloned into the pVSV backbone at position 5 with HiFi assembly. The full-length molecular clones termed ‘CDC pVSVΔG/VSV’, ‘CDC pVSVΔG/VSV-ZsG’, ‘CDC pVSVΔG/WEEV(IMP)’, and ‘CDC pVSVΔG/WEEV(McM)-ZsG’ were sequence verified by Oxford Nanopore sequencing (Plasmidsaurus, Louisville, KY, USA). Chimeric VSV/WEEV(IMP), VSV/VSV, and VSV/VSV-ZsG were rescued by transfecting BSR-T7/5 cells with six plasmids that encode for the VSV polymerase (L), nucleocapsid (N), phosphoprotein (P), glycoprotein (G), T7 RNA polymerase (Addgene, #65974), and the full-length molecular clone with transcription of antigenome under control of a T7 promotor. BSR T7/5 transfections were conducted with Lipofectamine 3000 (Invitrogen, Carlsbad, CA, USA) according to the manufacturer’s protocol. At 48 h post-transfection, BSR-T7/5 cells were overlaid on Vero cells and a successful rescue was determined by the appearance of cytopathic effect and reporter gene expression. Chimeric VSV/WEEV(McM)-ZsG was rescued with a similar approach using HEK293T cells instead of BSR-T7/5 cells and codon-optimized helper plasmids.

### 2.3. Virus Stocks

Chimeric VSV/WEEV, VSV/VSV, and parental WEEV were passaged in Vero cells with a low multiplicity of infection (MOI) of 0.01 for viral stock propagation. Virus stocks were propagated in Ye-Lah maintenance media [47] containing 50% 2X Ye-Lah media (Earle’s balanced salt solution without phenol red with 4% FBS, 0.2% 1000X gentamycin and 6.6% Ye-Lah solution (equal parts 2% (*w*/*v*) yeast extract solution to 10% (*w*/*v*) lactalbumin hydrolysate in water) and supplemented with 1% 10X Pen/Strep, 2 mM L-glutamine, 3% sodium bicarbonate (7.5%), and 43% cell culture water. Passage 2 of recovered VSV chimeras was verified by PCR screening with primers specific to the WEEV envelope polyprotein or VSV G using cDNA generated from extracted vRNA. Viral titers were determined by standard plaque assay [48], where the virus was serially diluted in BA-1 media (M-199 media supplemented with 0.05 M Tris-HCl (pH 7.5) containing 1% bovine serum albumin (BSA), 2 mM L-glutamine, 0.35 g/L sodium bicarbonate, and 100 U/mL penicillin, 100 mg/L streptomycin) [49].

### 2.4. Virus Purification

All viruses used in this study were purified on a glycerol (30%)–tartrate (45%) gradient by overnight ultracentrifugation (Beckman SW41 rotor; 26K RCF) as previously described [50]. The extracted virus band was pelleted in a subsequent 4-h ultracentrifugation (Beckman SW41 rotor; 37K RCF) spin and resuspended in TNE buffer (100 mM Tris, 10 mM, 1 mM EDTA, pH 8.0). The protein concentration of purified viral stocks was determined by a standard Bradford protein assay (BioRad, Berkeley, CA, USA) or a protein A280 reading on a Nanodrop spectrophotometer.

### 2.5. Immunoblotting

The protein content of purified viral stocks was determined by electrophoresis on a NuPAGE 4–12% Bis-Tris gel (Invitrogen, Carlsbad, CA, USA) under reduced conditions. Proteins were visualized with SimplyBlue safe stain (Invitrogen, Carlsbad, CA, USA). Protein band specificity was determined by immunoblotting of nitrocellulose (NC) membranes. Membranes were blocked overnight (4 °C) in phosphate buffered saline (PBS) with 5%(*v*/*v*) BSA or 10% (*v*/*v*) normal goat serum and probed with α-VSV monoclonal antibodies (MAbs) 8G5F11 (1:1000 in PBS) and 23H12 (1:1000 in PBS) (kind gifts from Dr. Douglas Lyles, Wake Forest School of Medicine), or α-WEEV mouse hyperimmune ascitic fluid (MHIAF) (CDC Arboviral Diseases Branch, Reference and Reagent Laboratory, Fort Collins, CO, USA) (1:1000 in blocking buffer). Goat α-mouse IgG-HRP conjugate (Jackson ImmunoResearch, West Grove, PA, USA) was diluted 1:5000 in PBS. Membranes were washed three times with PBS-T (PBS-0.1% Tween20) after incubation of the primary and secondary antibodies. Blots were developed with SuperSignal West Dura extended duration substrate (Thermo Scientific, Carlsbad, CA, USA) and chemiluminescence signals captured using an Azure Sapphire FL biomolecular imager.

### 2.6. Morphological Plaque Assessment

Approximately 50–100 plaque-forming units (PFUs) were incubated with Vero cells in a 6-well plate for 1 h at 37 °C prior to adding a 0.5% agarose in 2X Ye-Lah Media overlay. Plaques were visualized at 24-, 30-, 36-, 42-, and 48-h post-infection (hpi) using a crystal violet staining solution consisting of 40% (*v*/*v*) methanol and 0.28% (*w*/*v*) crystal violet [51]. The diameter and range of plaque diameters were approximated with a standard ruler.

### 2.7. Viral Kinetics

Wildtype and chimeric viral kinetics were assessed with an MOI of 0.001 in a multi-step growth curve in Vero cells with Ye-Lah maintenance media to buffer against media acidification. An additional flask containing only complete media (no virus) was used as a control (‘Mock’). After adsorption of the virus inoculum for 1 h at 37 °C, cells were washed three times, complete media was added, and then the initial sample (T = 0 h) was collected. At each time point, 5% of the cell supernatant was collected and replaced with fresh media. Viral titers were determined by standard plaque assay on Vero cells using a crystal violet staining solution as described above.

### 2.8. Plaque Reduction Neutralization Test (PRNT)

PRNTs were performed by incubating approximately 100 PFU of virus with equal volumes of sera at 37 °C for 1.5 h prior to adsorbing the virus–serum mix to Vero cells for 1 h at 37 °C with 5% CO_2_. Vero cells were inoculated in triplicate for each antibody–virus combination and a back-titration of input virus was performed. Both virus and sera were diluted 2-fold in BA-1 media. Cells were overlaid with a 0.5% agarose in 2X Ye-Lah Media, followed by either a second 0.5% agarose overlay the next day containing neutral red (Sigma, Medford, MA, USA) or crystal violet staining solution. A neutral red containing second overlay was used with VSV/VSV, VSV/WEEV(IMP), and WEEV(IMP) viruses and they were counted at both 2- and 3-days post-infection (dpi) to account for differences in growth kinetics between chimeric and parental viruses. Crystal violet staining solution was used with the VSV/VSV-ZsG, VSV/WEEV(McM), and WEEV(McM) due to the increased difficulty of reading the wildtype plaques. Cells were fixed at approximately 30, 32, and 42 h post-infection for each virus, respectively. The last serum dilution associated with a 90% reduction in plaques was used to assign PRNT_90_ titers for each trial.

### 2.9. Neutralization Curves

Percent neutralization was calculated for two independent trials using the input virus amount determined from a linear regression of the back-titration. Neutralization curves were generated with a 4-parameter non-linear regression dose–response fit with curve top and bottom constrained to 100 and 0, respectively, in GraphPad Prism (V10.0.3) from data generated across two independent trials (3 replicate wells per trial). The inhibitory concentration 90 (IC_90_) was calculated for each serum sample from the combined neutralization curve and is defined as the serum dilution attributed to a 90% reduction in PFUs.

### 2.10. Reporter Stability Assessment

Reporter stability with repeated passaging for VSV/VSV-ZsG and VSV/WEEV-ZsG was evaluated. P2 stocks were used in all the experiments described above. P3–P6 was generated by a low MOI infection (~0.01) in Vero cells. Viral stocks from each passage (P1–P6) were titrated by standard plaque assay on Vero cells as described above. At 24 hpi, live cell images of the reporter protein ZsGreen were captured with a Celigo instrument using the GREEN channel. Visual plaques formed at either 28 hpi (VSV/VSV-ZsG) or 33 hpi (VSV/WEEV-ZsG) from the same wells were compared to crystal violet staining as described above.

To further assess the propensity for reporter gene dropout, FACs analysis was performed to determine the presence of ZsGreen and VSV M within individual cells. Viral stocks from each passage were used to infect a T-25 flask of Vero cells at an MOI of 2.0, except for VSV/WEEV-ZsG P1, which was approximately 0.01 due to a low rescue titer. At 12 hpi (24 hpi for VSV/WEEV-ZsG P1), infected cells were lifted with trypsin and then fixed and permeabilized (BD Fix/Perm kit) prior to immunostaining with α-VSV M monoclonal antibody [23H12] (diluted 1:1000 in PBS-T) for 1.5 h at 4 °C, followed by an AlexaFluor647-conjugated goat anti-Mouse IgG (diluted 1:1000 in PBS-T) for 1 hr at 4 °C in the dark. Cells were washed three times with BD Fix/Perm wash buffer after incubation with each antibody and analyzed on a BD Accuri flow cytometer using 20,000–30,000 ‘single cell’ gated events. A gating strategy of ‘cells’ < ‘single cells’ < FITC-positive and AlexaFluor647-positive was imposed by using primary and secondary stained uninfected cells to set FITC and AlexaFluor647 gates.

Lastly, primers corresponding to the VSV genome, corresponding to the region flanking the inserted reporter gene (position 5), were used to evaluate reporter gene dropout with repeated passaging in tissue culture. Briefly, viral RNA was extracted from each stock to serve as a template in a one-step RT-PCR reaction (Titan One Tube RT-PCR). PCR amplicons were run on a 1% agarose gel in TBE buffer and stained with GelRed to visualize nucleic acid.

## 3. Results

### 3.1. Generation and Characterization of Chimeric VSV/WEEV

Chimeric VSV virions bearing WEEV E2-E1 envelope proteins (VSV/WEEV), or the native G glycoprotein (VSV/VSV) as a control, were produced to evaluate the performance of a neutralization assay employing infectious VSV/WEEV as opposed to authentic WEEV. Briefly, a full-length molecular clone encoding for VSV anti-genome (Indiana serogroup) was created and then modified to produce the WEEV (Group B3 strain IMP181 or Group A strain McMillan) envelope polyprotein (E3-E2-6K/TF-E1) instead of VSV G (Figure 1a,b). The VSV/WEEV (McMillan) chimera and the VSV/VSV chimera were engineered to also encode an extremely bright reporter protein, ZsGreen, to aid the development of reporter-based neutralization assays. All viruses (VSV/VSV, VSV/VSV-ZsG, VSV/WEEV (IMP), VSV/WEEV-ZsG (McM), and WEEV strain Imp-181) were recovered from DNA clones except for wildtype WEEV (strain McMillan), which was previously isolated from a human. Viruses were propagated once in Vero cells to generate stocks used for characterization.

The total protein content of purified chimeric VSV/WEEV demonstrated the presence of four VSV structural proteins (N, P, M, and L) and WEEV glycoproteins (E2 and E1) by SimplyBlue staining, whereas VSV/VSV chimeras contained all five native VSV structural proteins (Figure 1c,e). Importantly, immunoblotting for virus-specific proteins confirmed the incorporation of WEEV E2 and E1 and the absence of VSV G on chimeric VSV/WEEV, with only VSV-specific proteins detected on VSV/VSV particles and WEEV-specific proteins on WEEV particles (Figure 1d,f).

### 3.2. Viral Growth Kinetics of Chimeric VSV/WEEV Are Intermediate Between VSV/VSV and WEEV

A growth curve in Vero cells at a low MOI (0.001) was used to assess differences in virus propagation. Chimeric VSV/VSV multiplied the quickest and achieved the highest viral titers peaking at 7 × 10^8^ PFU/mL at 24 hpi (Figure 2a) or 9 × 10^8^ PFU/mL at 30 hpi for VSV/VSV-ZsG (Figure 2c). In contrast, authentic WEEV multiplied the slowest and had the lowest viral titers plateauing at 10^7^ PFU/mL at 42 hpi for both strains (Imp-181 and McMillan) (Figure 2a,c). Chimeric VSV/WEEV multiplied at a rate intermediate of VSV/VSV and WEEV, peaking at 30 hpi with a titer of 10^8^ PFU/mL regardless of the glycoprotein strain or the addition of a reporter gene (Figure 2a,c). The increased size of the envelope polyprotein (E3-E2-6K/TF-E1) of WEEV (3 kb) relative to VSV G (1.5 kb) or differences in viral entry kinetics due to glycoprotein-receptor binding may contribute to slightly lower growth of chimeric VSV/WEEV relative to VSV/VSV (Figure 2a,c).

Next, the plaque morphology of chimeric VSV/WEEV was examined relative to WEEV and VSV/VSV at 6-h intervals post-infection on Vero cells starting at 24 hpi. Chimeric VSV/VSV and VSV/WEEV plaques formed rapidly, with pinpoint plaques observed at 24 hpi (Figure 2b,d). By 30 hpi, the ideal plaque size for counting chimeric VSV/VSV was achieved, with plaque diameters averaging 1.3 mm using crystal violet staining solution (Table 1, Figure 2b,d). Chimeric VSV/WEEV plaques could be counted at 30 hpi with median sizes of 1.0 mm; however, plaques at 36 hpi are likely preferred with an average size of 1.5 mm (Table 1, Figure 2b,d). In comparison, authentic WEEV was delayed in plaque formation with average plaque diameters of 1.0 mm at 36 hpi and ideal plaque sizes of 1.5 mm occurring at 42 hpi (Table 1, Figure 2b,d). Morphologically, chimeric VSV/WEEV plaques display defined regular edges akin to VSV/VSV plaques as opposed to the diffuse and irregular borders of authentic WEEV (Figure 2b,d). While both WEEV (strain Imp-181) and WEEV (strain McMillan) produced countable plaques at 42 hpi, accurate counts were difficult due to heterogeneity in plaque size and shape (Table 1, Figure 2b,d). Consistent with the accelerated viral growth observed, chimeric VSV/WEEV produced countable plaques 6–12 h earlier than its parental WEEV counterpart.

### 3.3. Chimeric VSV/WEEV Performs Equivalently to Authentic WEEV in Neutralization Assays

A plaque reduction neutralization test (PRNT) was performed to evaluate the ability of sera to neutralize parental WEEV relative to chimeric VSV/WEEV using a panel of WEEV-specific, VSV-specific, and normal control sera. Overall, neutralization activity was analogous between parental WEEV and chimeric VSV/WEEV across serum specimens collected from mice, chicken, and horses, regardless of if the WEEV strain was from a Group B3 lineage (Imp-181, Figure 3) or a Group A lineage (McMillan, Figure 4) that also contained a reporter gene encoding for ZsGreen.

Chimeric VSV/WEEV exhibited notable increased sensitivity to neutralization with MHIAF raised against WEEV strain M2-958 (Figure 3c and Figure 4c). As expected, control sera from mice and horses did not produce dose-dependent neutralization (Figure 3f,i and Figure 4f,h). Critically, MHIAF raised against VSV (serogroup Indiana) did not neutralize VSV/WEEV or WEEV but strongly neutralized chimeric VSV/VSV across all serum dilutions tested (Figure 3f and Figure 4f). These data support that prior exposure to naturally circulating VSV in equines or humans does not interfere with the ability of chimeric VSV/WEEV to accurately discriminate WEEV neutralizing antibodies in a PRNT.

Both PRNT_90_, the last serum dilution tested, which achieved a 90% reduction in plaques, and IC_90_, the calculated dilution at which a 90% inhibition of plaques occurs, were determined from neutralization assays conducted across 2-fold serial dilutions of sera against either chimeric VSV/WEEV or authentic WEEV. PRNT_90_ titers were either equivalent or higher for chimeric VSV/WEEV than authentic WEEV (Table 2 and Table 3). For example, two anti-WEEV MHIAF serum specimens (Fleming^1^ and Fleming^2^) did not achieve the PRNT_90_ threshold with parental WEEV (Imp-181), whereas chimeric VSV/WEEV (Imp-181) had PRNT_90_ reciprocal dilution values of 20 or 80 and 80 or 40, for each trial, respectively (Table 2). Four of the seven sera tested (MHIAF Y62-33, MHIAF M2-958, MHIAF Fleming, and Horse Fleming) had higher IC_90_ titers for chimeric VSV/WEEV (Imp-181) than WEEV (Imp-181) with nonoverlapping 95% confidence intervals (CIs) (bolded in Table 2). Two serum specimens had higher IC_90_ values with overlapping 95% CIs for chimeric VSV/WEEV compared to WEEV (MHIAF Fleming^1^: 62 [CI:43–92] vs. 27 [CI:15–48] and Chicken unknown: 128 [CI:87–187] vs. 60 [CI:40–91], respectively). While the calculated IC_90_ for the wildtype WEEV (Imp-181) with MHIAF (pooled) was higher than chimeric VSV/WEEV (Imp-181), values are within a 2-fold dilution of each other, the 95% CIs overlap, and the PRNT_90_ values were equivalent between each trial (Table 2). IC_90_ values for chimeric VSV/WEEV-ZsG (McMillan) were higher than authentic WEEV (McMillan) for each specimen tested, with nonoverlapping CIs for each specimen except MHAIF (pooled) (Table 3). PRNT_90_ between Imp-181 and McMillan strains had values within a single two-fold dilution. When IC_90_ values of VSV/WEEV strains Imperial and McMillian were compared to the corresponding parental strains using the extra sum-of-squares *F*-test, significant differences were observed for all sera tested except for MHIAF (pooled), which showed no significant difference in IC_90_ fits for either the Imperial strain (*p* = 0.2880) or the McMillian strain (*p* = 0.0760) and corresponding VSV/WEEV chimeras. Both the calculated PRNT_90_ and IC_90_ values support the utility of chimeric VSV/WEEV as a suitable surrogate for WEEV in a diagnostic PRNT.

### 3.4. Reporter Gene, ZsGreen, Remains Stable with Repeated Chimeric rVSV Passaging in Tissue Culture

The bright green fluorescent protein, ZsGreen (~700 bp), was cloned into the full-length molecular clone of VSV to generate replication-competent recombinant VSV particles that also produce ZsGreen in the cytoplasm of infected cells as a reporter-based readout for the development of high-throughput neutralization assays. Reporter-based assays have faster turnaround times than traditional plaque assays, which require multiple rounds of virus replication and a laboratorian to manually count plaques. Other chimeric systems that employ reporter genes have noted reporter gene dropout with repeated viral passaging [52]. While only P2 stocks were used in the above characterizations, the stability of reporter protein expression, ZsGreen, with repeated passaging of virus on Vero cells was assessed to confirm the utility of reporter-based readouts with VSV/VSV-ZsG and VSV/WEEV-ZsG chimeras.

First, P1-P6 viral stocks were titrated on Vero cells. Wells were scanned with a live cell imager (Celigo) to capture images of ZsGreen foci ahead of cell fixation and staining with crystal violet. Both VSV/VSV-ZsG and VSV/WEEV-ZsG retained nearly 100% accurate counts between ZsGreen foci to visible plaques for all six passages (Figure 5a,b and Appendix A). Additionally, viral stocks from each passage were used to infect Vero cells with a high MOI (2) and cells were fixed at 12 hpi. FACs analysis confirmed ZsGreen stability across all six passages of virus stock with no indications of a loss of ZsGreen expression for either VSV/VSV-ZsG or VSV/WEEV-ZsG (Figure 5c and Appendix A). Roughly 10% of Vero cells stained negative for VSV M (AlexaFlour-647; APC-A), except for P1 of VSV/WEEV-ZsG, but were still nearly 100% ZsGreen positive (FITC-A) (Figure 5c). This could be due to either incomplete staining or differences in the sensitivity of detection between ZsGreen and AlexaFluor-647. The P1 stock of VSV/WEEV-ZsG had a low titer, and thus multiple rounds of replication occurred prior to cell fixation at 24 hpi, with cells displaying extensive cytopathic effect at the time of harvest.

Next, viral RNA was extracted from each viral stock and used as a template in a one-step RT-PCR reaction amplifying the genomic region flanking position 5 (Figure 1). Both VSV/VSV-ZsG and VSV/WEEV-ZsG retained the complete genomic region encoding for ZsGreen compared to the chimeric viruses VSV/VSV and VSV/WEEV (Imp-181), which contain the priming sites but lack the inserted reporter gene, ‘ZsG (−)’ (Figure 5d and Appendix A). Overall, these data support that ZsGreen reporter expression is stable in these replication-competent chimeric viruses and that ZsGreen foci can recapitulate counts obtained from visual plaque-forming units from low passage stocks. Reporter-based assays allow for the future development of automated high-throughput platforms with more rapid readouts of neutralization tests compared to the traditional PRNT.

## 4. Discussion

WEEV has caused significant epizootics in the Americas since its isolation in 1930. Most regions with enzootic WEEV circulation do not conduct systematic surveillance, precluding the detection of increased prevalence in enzootic cycles that often signifies increased epizootic risk. A contributing factor to the lack of surveillance and diagnostic capacity in the region is the limited number of local facilities equipped for BSL-3 biocontainment. Despite the drawbacks of the PRNT, it remains the reference standard over alternative serological methodologies (e.g., ELISA, MIA) due to its high-test specificity. There is a clear need for improved diagnostic reagents to expand testing capacity for WEEV in the Americas. These data support the utilization of replication-competent chimeric VSV particles bearing WEEV E2/E1 spikes as a surrogate for authentic WEEV in a PRNT. Using chimeric VSV/WEEV in PRNTs instead of authentic WEEV has several advantages including (1) faster plaque development, (2) improved ease of counting due to a consistent plaque morphology with defined regular edges, (3) decreased health risk to testing personnel, (4) ability to develop high-throughput reporter-based neutralization tests and critically, (5) the ability to perform testing at a lower biosafety containment level.

Chimeric VSV/WEEV exhibits several advantageous features compared to authentic WEEV in terms of growth kinetics and plaque morphology. Not only could chimeric VSV/WEEV plaques be counted sooner (at 32 hpi using crystal violet staining), but counts were easier to perform with less ambiguity within and between adjacent plaques. This feature lends to more reliable neutralization measurements with the potential for decreased variation between laboratorians enumerating plaques.

We found chimeric VSV/WEEV tended to have higher neutralizing titers than parental WEEV across our serum panel. This may be due to differences in the abundance and arrangement of trimeric E2/E1 spikes on the virion. VSV particles are bullet-shaped particles that are approximately 70 nm by 200 nm [53], whereas alphavirus virions are spherical with a diameter of approximately 70 nm and contain 80 trimeric spikes of E2/E1 heterodimers [17]. Differences in the arrangement and abundance of spike complexes on the VSV bullet-shaped virion may enable some antibodies to readily access certain epitopes that may be cryptic on the authentic viral particle. While VSV can readily incorporate foreign proteins into the viral particle, the efficiency of incorporation and arrangement of E2/E1 spikes on recombinant VSV vectors is currently unknown and should be further characterized. Interestingly, SINV-based new world alphaviruses, which theoretically should display E2/E1 spikes with congruent arrangements and abundance as wildtype virus, were found to be less sensitive to neutralization than authentic EEEV counterparts, albeit analogous for clinical detection [35].

Further evaluation with a panel of human and equine sera from the 2023—2024 epizootic should be conducted to confirm neutralization of chimeric VSV/WEEV in a diagnostic PRNT with antibodies produced from circulating South American strains. Prototypical strains of WEEV, such as McMillan and Flemming, are classified as members of the ‘Group A’ lineage. Over time, the circulation of Group A strains was displaced by Group B strains, with lineages further subclassified into Group B1, B2, and B3 [21]. Currently, only Group B3 strains circulate in North America. One VSV/WEEV chimera used in this study was constructed with the E3-E1 polyprotein from a Group B3 strain (Imp-181) isolated in 2005 from a mosquito pool in California. The South American isolates from the 2023–2024 outbreak are genetically distinct from the North American isolates and have a proposed classification as a novel ‘Group C’ lineage [1,2,3]. The South American 2023–2024 isolates are closely related to a 1958 isolate CBA87 from Argentina [2]. However, few complete genome sequences exist from isolates of prior South American epizootics, limiting inferences of evolutionary history. WEEV neutralizing antibodies predominantly recognize E2 (facilitates attachment) due to the pre-fusion spike conformation [54,55], but can also be elicited to cryptic broadly-conserved E1 epitopes (facilitates fusion) [56,57]. E2 and E1 amino acid residues of isolates from the 2023–2024 isolates closely match (≥96.7%) lineage Group A isolates (Appendix A). The Group A-specific sera used in this study were able to efficiently neutralize chimeric VSV/WEEV bearing E2/E1 heterotrimeric spikes from both a Group B3 Imp-181 isolate and a Group A McMillan isolate, suggesting that sera from novel Group C isolates would also neutralize chimeric VSV/WEEV despite the lineage differences. The absence of human WEE cases within the United States over the last two decades has precluded our ability to conduct a more thorough assessment of chimeric VSV/WEEV with relevant clinical specimens.

The use of attenuated virus chimeras as opposed to authentic pathogenic viruses represents significant advancements in safety, especially for viruses that have been associated with laboratory-acquired infections due to procedural aerosols [58]. We did not directly assess the attenuation of chimeric VSV/WEEV relative to authentic WEEV in animal models. Thus, while the evaluation of recombinant VSV/VEEV or VSV/EEEV or SINV/encephalitic alphavirus has suggested its safe use at lower biocontainment levels (BSL-2) [25,42], further testing should be performed to support these biosafety reductions. Future work should aim to build on the reporter framework to develop high-throughput diagnostic neutralization tests. Improvements in diagnostic reagents have the potential to significantly enhance arboviral detection in the Americas, ultimately strengthening public health efforts to respond and prevent outbreaks of WEEV and other emerging arboviruses.

## Figures and Tables

**Figure 1 viruses-17-01067-f001:**
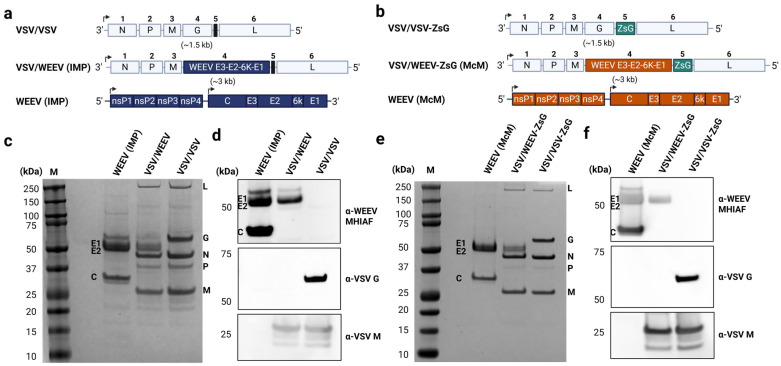
Characterization of chimeric VSV/VSV, VSV/WEEV, and parental WEEV virions. (**a**) Genome organization for chimeric VSV/VSV, VSV/WEEV (IMP), and parental WEEV (IMP) viruses, (**b**) Genomic organization for chimeric viruses encoding the ZsGreen (ZsG) reporter protein, VSV/VSV-ZsG, VSV/WEEV-ZsG (McM), and wildtype authentic WEEV McMillan (McM) strain. Chimeric VSV/VSV has a negative-sense single-stranded RNA genome which encodes five structural proteins (N, P, M, G, and L). Chimeric VSV/WEEV was engineered to express the envelope polyprotein of WEEV (E3-E2-6K-E1) instead of the cognate VSV-G. WEEV has a positive-sense RNA genome that is divided into two open reading frames (ORFs). (**c**,**e**) Total protein content in purified viral stocks by SimplyBlue staining. ‘M’ indicates the protein marker, with the structural proteins for WEEV indicated on the left and for VSV on the right. (**d**,**f**) Protein specificity was determined by immunoblotting probed with either α-WEEV MHIAF, α-VSV G, or α-VSV M and detection of chemiluminescent signals.

**Figure 2 viruses-17-01067-f002:**
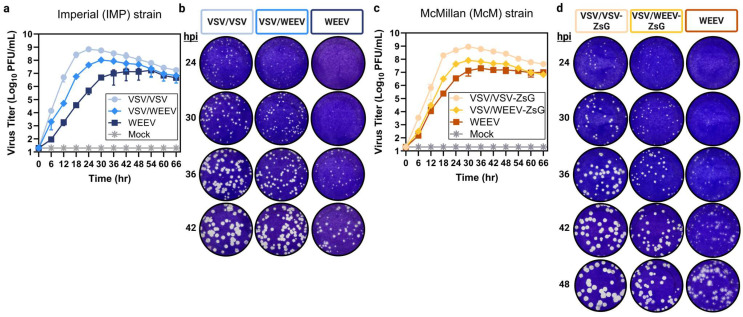
Virus kinetics and plaque morphology of chimeric VSV/VSV, VSV/WEEV, and parental WEEV. (**a**,**c**) Viral kinetics of chimeric VSV/VSV (light blue or light orange), chimeric VSV/WEEV (medium blue or medium orange), and parental WEEV (dark blue or dark orange) was assessed in Vero cells using an MOI of 0.001. A negative control (media only) ‘Mock’ (grey) was included for both experiments. Culture supernatants were collected every 6 h and virus titers were determined by plaque assay in triplicate from two independent trials. (**b**,**d**) Plaques formed from virus infection (~50–100 PFU) with either chimeric VSV/VSV, VSV/WEEV, or authentic WEEV were morphologically assessed at 24, 30, 36, 42, and 48 hpi (later only in **d**) using crystal violet staining solution.

**Figure 3 viruses-17-01067-f003:**
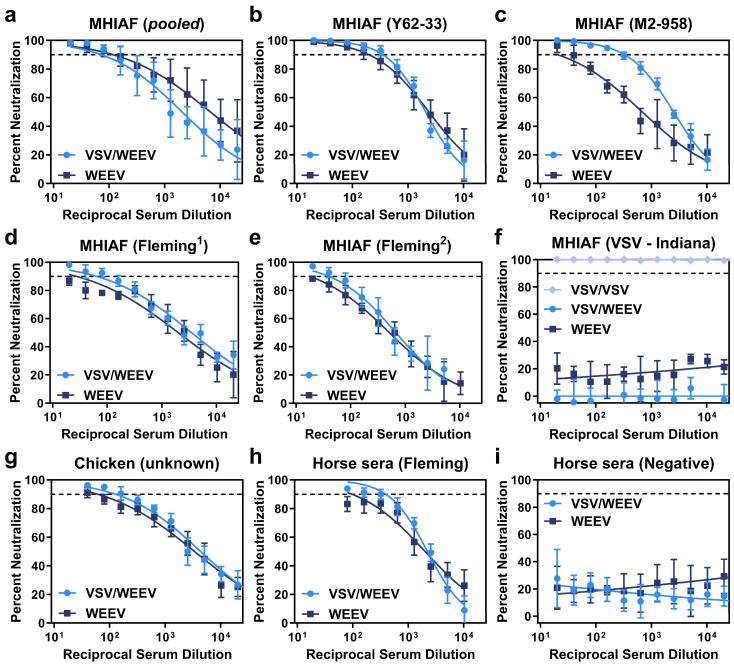
Neutralization curves of chimeric VSV/WEEV and parental WEEV strain Imp-181. Neutralization curves were fitted with a non-linear regression dose–response curve to the percent neutralization generated from PRNT data performed with immune sera over a range of two-fold serial dilutions. Percent neutralization was normalized to the input virus, calculated from a linear regression of the back-titration of the virus inoculum. The IC_90_ threshold is indicated with a dashed line. Percent neutralization of chimeric VSV/WEEV to parental WEEV was assessed with a panel of sera corresponding to MHIAF raised against strains of WEEV including pooled sera (**a**), Y62-33 (**b**), M2-958 (**c**), Fleming^1^ (**d**), and Fleming^2^ (**e**) or raised again VSV strain Lab (serogroup Indiana) (**f**). Anti-WEEV positive chicken sera (**g**). Additional sera from equines(horse), either anti-WEEV Fleming positive (**h**) or normal control sera (**i**), were tested for neutralization of VSV/WEEV or parental WEEV.

**Figure 4 viruses-17-01067-f004:**
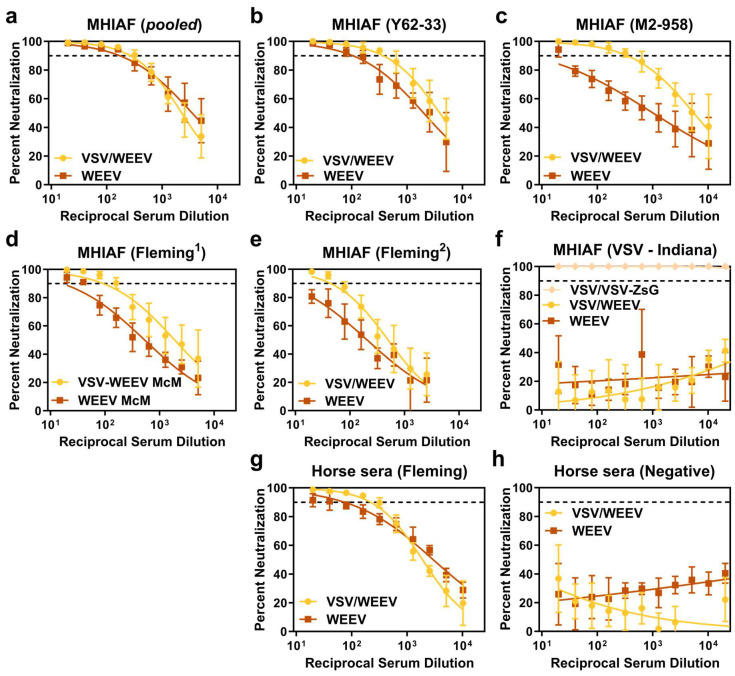
Neutralization curves of chimeric VSV/WEEV-ZsG and parental WEEV strain McMillan. Neutralization curves were fitted with a non-linear regression dose–response curve to the percent neutralization generated from PRNT data performed with immune sera over a range of two-fold serial dilutions. Percent neutralization was normalized to the input virus, calculated from a linear regression of the back-titration of the virus inoculum. The IC_90_ threshold is indicated with a dashed line. Percent neutralization of chimeric VSV/WEEV-ZsG to parental WEEV was assessed with a panel of sera corresponding to MHIAF raised against strains of WEEV including pooled sera (**a**), Y62-33 (**b**), M2-958 (**c**), Fleming_1_ (**d**), and Fleming_2_ (**e**) or raised again VSV strain Lab (serogroup Indiana) (**f**). Additional sera from equines(horse), either anti-WEEV Fleming positive (**g**) or normal control sera (**h**), were tested for neutralization of VSV/WEEV or parental WEEV.

**Figure 5 viruses-17-01067-f005:**
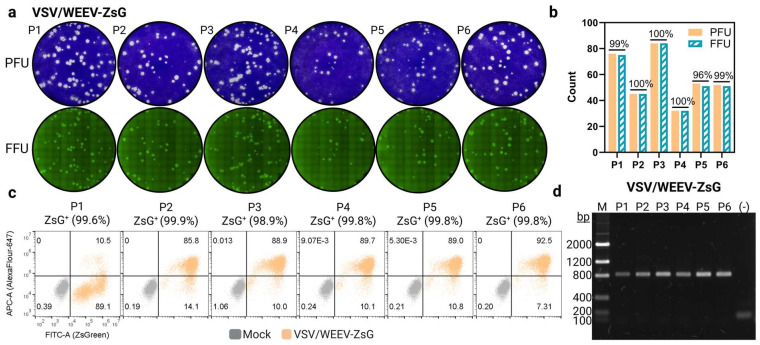
ZsGreen stability in chimeric VSV/WEEV-ZsG with repeated passaging in Vero cells (**a**) Comparison of PFUs (visual) to FFU (ZsGreen) in Vero cells across six passages (**b**) Enumeration of PFUs to FFU in each well (**c**), FACs analysis of ZsGreen expression (MFI) versus cells collected at 12 hpi stained for VSV-M (AlexaFour-647) in Vero cells infected with a high MOI (1–2) except P1 (MOI = 0.01, collected 24 hpi) of VSV/WEEV-ZsG (light orange) using stocks from each passage. Cells were gated for the bulk population, then singlets, with FITC-A and APC-A gates selected from uninfected Vero cells stained with both primary and secondary antibodies (grey population) (**d**), PCR amplicons of genomic position 5 using a one-step RT-PCR reaction with viral RNA from P1-P6 stocks as template and chimeric VSV/WEEV without ZsGreen as a negative (−) control.

**Table 1 viruses-17-01067-t001:** Mean plaque diameter and plaque size ranges of chimeric and parental viruses.

Virus	Mean Plaque Diameter (Range) in mm				
	24 hpi	30 hpi	36 hpi	42 hpi	48 hpi
VSV/VSV	1 (<0.5–1.0)	1.3 (<0.5–2.0)	2.0 (0.5–2.5)	2.5 (0.5–3.0)	---
VSV/WEEV (IMP)	0.5 (<0.5–1.0)	1.0 (<0.5–1.3)	1.5 (0.5–2.0)	2.0 (1.0–2.5)	---
WEEV (IMP)	Not detectable	<0.5 (<0.5–0.75)	0.75 (<0.5–1.0)	1.5 (<0.5–2.0)	---
VSV/VSV-ZsG	1.0 (<0.5–1.0)	1.3 (<0.5–1.5)	2.0 (0.5–2.5)	2.5 (1.5–3.0)	3.0 (2.0–3.5)
VSV/WEEV-ZsG (McM)	0.5 (<0.5–1.0)	1.0 (<0.5–1.3)	1.5 (0.5–2.0)	2.0 (1.0–2.0)	2.5 (1.0–2.5)
WEEV (McM)	Not detectable	<0.5 (<0.50)	1.0 (<0.5–1.3)	1.5 (<0.5–2.0)	2.0 (<0.5–3.0)

**Table 2 viruses-17-01067-t002:** Neuralization assay metrics for surrogate chimeric VSV/WEEV and parental WEEV strain Imp-181.

Antibody			IC_90_ ^1^		PRNT_90_ ^2^			
Specificity	Source	Isolate	VSV/WEEV	WEEV	VSV/WEEV		WEEV	
			Mean (95% CI)	Mean (95% CI)	Trial 1	Trial 2	Trial 1	Trial 2
WEEV	mouse	pooled ^3^	72 (40–130)	118 (57–246)	160	80	160	80
WEEV	mouse	Y62-33	354 (289–433)	198 (137–284)	160	320	160	80
WEEV	mouse	M2-958	328 (280–385)	20 (12–35)	160	320	20	40
WEEV	mouse	Fleming_1_	62 (43–92)	27 (15–48)	20	80	<20	<20
WEEV	mouse	Fleming_2_	40 (26–62)	<20	80	40	<20	<20
WEEV	chicken	unknown	128 (87–187)	60 (40–91)	80	160	40	<40
WEEV	horse	Fleming	393 (313–494)	97 (59–158)	320	320	20	20
VSV	mouse	Indiana	<20	<20	<20	<20	<20	<20
------	horse	-------	<20	<20	<20	<20	<20	<20

^1^ Inhibitory concentration 90 (IC_90_): values shown as the reciprocal of the dilution of sera yielding at least 90% reduction in percent neutralization. ^2^ Plaque reduction neutralization test 90 (PRNT_90_): values shown as the reciprocal of the dilution of sera yielding at least 90% reduction of the input number of PFU.^3^ Pooled MHIAF consisting of Fleming (lot: M29781), CBA (lot: M18516), R43738 (lot: M20829). PRNT_90_ titer ≥ 20; interpret result as positive, with <20 as negative or below detection limit.

**Table 3 viruses-17-01067-t003:** Neuralization assay metrics for surrogate chimeric VSV/WEEV-ZsG and parental WEEV strain McMillan.

Antibody			IC_90_ ^1^		PRNT_90_ ^2^			
Specificity	Source	Isolate	VSV/WEEV	WEEV	VSV/WEEV		WEEV	
			Mean (95% CI)	Mean (95% CI)	Trial 1	Trial 2	Trial 1	Trial 2
WEEV	mouse	pooled ^3^	267 (200–344)	172 (109–258)	320	160	160	160
WEEV	mouse	Y62-33	372 (256–520)	105 (63–165)	640	320	160	80
WEEV	mouse	M2-958	348 (224–517)	<20	320	160	20	20
WEEV	mouse	Fleming_1_	93 (53–149)	<20	160	80	40	40
WEEV	mouse	Fleming_2_	44 (27–67)	<20	40	40	<20	<20
WEEV	horse	Fleming	219 (173–272)	74 (52–103)	320	160	40	<20
VSV	mouse	Indiana	<20	<20	<20	<20	<20	<20
------	horse	-------	<20	<20	<20	<20	<20	<20

^1^ Inhibitory concentration 90 (IC_90_): values shown as the reciprocal of the dilution of sera yielding at least 90% reduction in percent neutralization. ^2^ Plaque reduction neutralization test 90 (PRNT_90_): values shown as the reciprocal of the dilution of sera yielding at least 90% reduction of the input number of PFU. ^3^ Pooled MHIAF consisting of Fleming (lot: M29781), CBA (lot: M18516), R43738 (lot: M20829). PRNT_90_ titer ≥ 20; interpret result as positive, with <20 as negative or below detection limit.

## Data Availability

The original contributions presented in this study are included in the article/Appendix A. Further inquiries can be directed to the corresponding author.

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
