# Peer review of "Chimeric Vesicular Stomatitis Virus Bearing Western Equine Encephalitis Virus Envelope Proteins E2-E1 Is a Suitable Surrogate for Western Equine Encephalitis Virus in a Plaque Reduction Neutralization Test"

_viruses, 2025, doi:10.3390/v17081067_

Round 1

Reviewer 1 Report

Comments and Suggestions for Authors

This is a nice manuscript. I consider it to be well-written, methodologically robust, and scientifically sound. The study is clearly presented and the data support the conclusions drawn by the authors.

I am particularly impressed by the biosafety aspects of the work. The development and validation of the chimeric VSV/WEEV system represents an important advance, as it significantly reduces the risks associated with working with live WEEV while maintaining diagnostic utility. This innovation will greatly facilitate the expansion of safe and effective PRNT capacity, especially in regions without ready access to BSL‑3 facilities.

I would encourage the authors to consider adding a short paragraph discussing alternative serological approaches for WEEV. In particular, it would be helpful to briefly address why PRNT—despite being a well‑established but labor‑intensive and live‑virus‑dependent method—remains the reference standard. A comparison with simpler assays that do not require live virus and are less resource‑intensive (for example, ELISAs or other emerging platforms) would provide useful context for readers and further highlight the significance of their chimeric VSV/WEEV approach.

Author Response

Comments 1: “I would encourage the authors to consider adding a short paragraph discussing alternative serological approaches for WEEV. In particular, it would be helpful to briefly address why PRNT—despite being a well-established but labor intensive and live virus dependent method—remains the reference standard. A comparison with simpler assays that do not require live virus and are less resource intensive (for example, ELISAs or other emerging platforms) would provide useful context for readers and further highlight the significance of their chimeric VSV/WEEV approach.”

Response 1: We thank the reviewer for this comment and agree this is an important aspect to emphasize. We briefly mention the use of the CDC MAC-ELISA in the introduction (lines 70-78). However, we have modified the following text in the introduction (lines 78-79) to further emphasize this point. “Due to potential cross-reactivity associated with testing methods such as the MAC-ELISA which does not require the use of live virus, confirmatory testing is performed with the highly specific plaque reduction neutralization test (PRNT) for detection of anti-WEEV neutralizing antibodies (NAbs).”

In addition, the following text was added to the discussion (lines 478-479). “Despite the drawbacks of the PRNT, it remains the reference standard over alternative serological methodologies (e.g. ELISA, MIA) due to its high-test specificity.”

Reviewer 2 Report

Comments and Suggestions for Authors

This manuscript describes the construction and implementation of two VSV/WEEV chimeras, one of which contains a fluorescent marker. The authors address the suitability of these VSV chimeras as surrogates within the context of antibody neutralization assays, as well as the stability of the fluorescent marker for the future implementation of high throughput assays. Overall, the work was performed thoroughly and presented well. It is especially relevant given the resurgence of WEEV in South America. I have only minor concerns which can be addressed without the need for further experiments and which I believe will strengthen the manuscript for publication:

  1. One of the major advantages described in this manuscript is the ability of the VSV/WEEV chimera to facilitate neutralizing antibody tests without the need for BSL-3 facilities. VSV chimeras are well described as platforms for surrogate assays and even vaccines that can be utilized at lower biosafety levels than the parental/donor virus, and the assertion is scientifically reasonable. However, the general track record of VSV does not mean that new safety testing is not needed from a regulatory perspective to demonstrate that a novel donor/backbone combination retains an attenuated phenotype. This point needs to be elaborated upon, either by stating the means by which institutional permission was obtained for handling at a lower biosafety level (and ideally providing any applicable data as a supplement) or by explicitly stating that this manuscript does not address any potential biosafety concerns related to the use of VSV/WEEV in the BSL-2. A brief (1-2 sentence) mention in the drawbacks and future directions portion of the discussion would sufficient, but it does need to be addressed.
  2. You specify that the VSV backbone is the Indiana serogroup (line 136). Is strain-level data available? If so, please include that information as well.
  3. The rescue of VSV/WEEV(IMP), VSV/VSV, and VSV/VSV-ZsG is described as taking place in BSR-T7/5 cells (lines 150-152), which stably express T7 RNA polymerase (line 116), but also state that a plasmid encoding T7 RNA polymerase was transfected into the BSR-T7/5 cells during the recovery process (line 153). Can you clarify why the T7 RNA polymerase-encoding plasmid had to be transfected into cells that already stably express the protein?
  4. Lines 158-160, is there a reason why VSV/WEEV(McM)-ZsG was recovered in a different cell line?
  5. The descriptive comparisons of the neutralization curves in results section 3.3 would be bolstered by performing a statistical comparison of the two curves (this can be done in GraphPad with the Compare options set to either see whether the two curves in a given panel are statistically identical or to compare specific parameters such as IC90).

Author Response

Comment 1: “One of the major advantages described in this manuscript is the ability of the VSV/WEEV chimera to facilitate neutralizing antibody tests without the need for BSL-3 facilities. VSV chimeras are well described as platforms for surrogate assays and even vaccines that can be utilized at lower biosafety levels than the parental/donor virus, and the assertion is scientifically reasonable. However, the general track record of VSV does not mean that new safety testing is not needed from a regulatory perspective to demonstrate that a novel donor/backbone combination retains an attenuated phenotype. This point needs to be elaborated upon, either by stating the means by which institutional permission was obtained for handling at a lower biosafety level (and ideally providing any applicable data as a supplement) or by explicitly stating that this manuscript does not address any potential biosafety concerns related to the use of VSV/WEEV in the BSL-2. A brief (1-2 sentence) mention in the drawbacks and future directions portion of the discussion would sufficient, but it does need to be addressed.”

Response 1: We thank the reviewer for this comment and agree this is an essential point to include. The evaluation of chimeric equine encephalitic viral-vectors (specifically SINV-based and VSV-based) has been performed by others (see references below). We agree that more extensive safety and immunogenicity testing should be performed prior to advocating for its use as vaccine vector or widespread diagnostic tool. As we did not perform animal studies to explicitly test this, we have included the below text to address this concern in the discussion (lines 534-538). “We did not directly assess the attenuation of chimeric VSV/WEEV relative to authentic WEEV in animal models. Thus, while the evaluation of recombinant VSV/VEEV or VSV/EEEV or SINV/encephalitic alphavirus has suggested its safe use at lower biocontainment levels (BSL-2)(1,2), further testing should be performed to support these biosafety reductions.”

Nasar, F., et al. (2017). "Recombinant Isfahan Virus and Vesicular Stomatitis Virus Vaccine Vectors Provide Durable, Multivalent, Single-Dose Protection against Lethal Alphavirus Challenge." J Virol 91(8).

Atasheva, S., et al. (2009). "Chimeric alphavirus vaccine candidates protect mice from intranasal challenge with western equine encephalitis virus." Vaccine 27(32): 4309-4319. 

Comment 2: “You specify that the VSV backbone is the Indiana serogroup (line 136). Is strain-level data available? If so, please include that information as well.”

Response 2: Due to the synthetic nature of the full-length molecular clone the strain is not homogenous. VSV genes were selected from both the Mudd-Summers and San Juan isolates. Thus, the isolate is not specified in the text since it is a conglomerate of isolates from the Indiana serogroup. To emphasize this point we added the following text to line 137. … “composed from strains Mudd-Summers and San Juan”

Comment 3: “The rescue of VSV/WEEV(IMP), VSV/VSV, and VSV/VSV-ZsG is described as taking place in BSR-T7/5 cells (lines 150-152), which stably express T7 RNA polymerase (line 116), but also state that a plasmid encoding T7 RNA polymerase was transfected into the BSR-T7/5 cells during the recovery process (line 153). Can you clarify why the T7 RNA polymerase-encoding plasmid had to be transfected into cells that already stably express the protein?”

Response 3: We thank the reviewer for pointing this out. The authors have previously rescued rVSV from BSR-T7/5 cells without the addition of the T7 RNA polymerase-encoding plasmid (see references below). While not essential to rVSV rescue, we suspect that the extra plasmid-encoded T7 RNA polymerase supplied could help boost rescue efficiencies. In non-quantitative assessments of T7 RNA polymerase activity using a plasmid encoding eGFP under the control of a T7promoter transfected into BSR-T7/5 cells, we observe brighter eGFP signals in BSR-T7/5 transfected with additional T7 RNA polymerase than BSR-T7/5 cells alone.

Reyes Ballista, J. M., et al. (2023). "Chikungunya virus entry and infectivity is primarily facilitated through cell line dependent attachment factors in mammalian and mosquito cells." Front Cell Dev Biol 11: 1085913.

Lay Mendoza, M. F., et al. (2020). "Monitoring Viral Entry in Real-Time Using a Luciferase Recombinant Vesicular Stomatitis Virus Producing SARS-CoV-2, EBOV, LASV, CHIKV, and VSV Glycoproteins." Viruses 12(12).

Acciani, M. D., et al. (2021). "Ebola Virus Requires Phosphatidylserine Scrambling Activity for Efficient Budding and Optimal Infectivity." J Virol 95(20): e0116521.

Comment 4: “Lines 158-160, is there a reason why VSV/WEEV(McM)-ZsG was recovered in a different cell line?”

Response 4: We commend the reviewer for pointing out this detail. Several rescue attempts of VSV/WEEV(McM)-ZsG using the same protocol as the other rescues in this manuscript failed. Thus, we performed an alternative rescue protocol (adapted from reference below) with the exact same full-length molecular clone prep several months later that employed codon-optimized helper plasmids and 293T cells and obtained an almost a 100% rescue efficiency. We did not go back and try this alternative protocol in BHKT7/5 cells to determine if the difference in rescue success could be attributed to the new helper plasmids or the cell line employed as we had used the remainder of the original full-length clone prep. In the literature, there is a wide variety of approaches researchers employ to rescue rVSV particles.

Drake, M. J., et al. (2017). "A role for glycolipid biosynthesis in severe fever with thrombocytopenia syndrome virus entry." PLoS Pathog 13(4): e1006316.

Comment 5: “The descriptive comparisons of the neutralization curves in results section 3.3 would be bolstered by performing a statistical comparison of the two curves (this can be done in GraphPad with the Compare options set to either see whether the two curves in a given panel are statistically identical or to compare specific parameters such as IC90).”

Response 5: We thank the reviewers for this suggestion to strengthen our conclusions. We have added the results of a comparative analysis of IC90s for the VSV/WEEV chimeras and the corresponding parental strain (lines 378-382). “When IC₉₀ values of VSV/WEEV strains Imperial and McMillian were compared to the corresponding parental strains using the extra sum-of-squares F-test, significant differences were observed for all sera tested except for MHIAF (pooled), which showed no significant difference in IC₉₀ fits for either the Imperial strain (p = 0.2880) or the McMillian strain (p = 0.0760) and corresponding VSV/WEEV chimeras.”